# Domain Adversarial Representation Learning for Data Independent Defenses against Poisoning Attacks

## Abstract

Understanding the worst case loss of a defense against a determined attack is important to evaluate the robustness of a particular classification algorithm to data poisoning attacks. Even though there are many methods for defending against attacks, they are dependent on the separability of the dataset representation. We pose this as a domain adaptation problem and learn a function in an adversarial setting to transform a dataset from a source domain to a target domain which has an established separability of clusters. The defenses thus obtained in the target domain show tighter upper bounds as compared to those in the source domain.

## 1 Introduction

With the increasing applications of Machine Learning to several domains, there is an increasing thrust on developing reliable systems which are robust to data poisoning attacks. Such defenses fall under two categories (i) Fixed Defenses, wherein there is the availability of clean dataset on which the training is performed (ii) Data-dependant defenses, wherein we only have access to a poisoned dataset. There are realistic use cases of fixed defenses, for example, a cloud based application could be trained on a clean dataset but is exposed to attacks during inference and this work focuses on fixed defenses.

Even though there are many defenses proposed for potential attacks, not much study has been done on the worst-case test loss for the dataset. The paper (Steinhardt et al., 2017) proposes a method to determine the upper bound on the test-loss for the proposed defenses. They achieve this by constructing feasible areas in the high dimensional space which aids to remove outliers as well as bound the maximum loss which can be incurred.

**Motivation**: The authors of Steinhardt et. al. mention that the proposed defenses are highly data-dependent and work only when the positive and negative clusters are well separated as in the case of MNIST 1-7 dataset (author). The performance of these defenses degrades dramatically when there isn't a clear separation in the clusters. Borrowing idea from Domain Adaptation we wanted to transform the IMDB dataset to the domain of MNIST 1-7. To additionally test the robustness of the proposed method, we have also trained a function to transform the SVHN dataset(shown in Fig 1) which doesn't have a well-defined class boundary and have c

## 2 Proposed Methodology

### 2.1 Dataset as Samples from a Probability Function

Consider a dataset $D$ consisting of $c$ classes. Each of the example is an $n$-dimensional vector which can be interpreted as a point in the $n$-dimensional vector space $\Re^n$. Without any loss of generality we can define the examples from each class as samples from a class probability distribution over $\Re^n$. Computing the exact probability distribution function $P_D^c$, for every class c is more often than not, intractable. Hence, we will be learning function which draws samples from an approximation of this function. In the following sections, we will be considering this abstraction in our formulation.

## 2.2 PROBLEM DEFINITION

Consider a classification task from the input $x \in X$ to an output $y \in Y$. In this work we restrict ourselves to the scenario where $X = \Re^d$ as adversarial domain adaptation works only on continuous samples. For the sake of convenience, we also assume $Y \in \{1, -1\}$ although the following analysis is valid for an arbitrary number of classes.

We have two clean datasets, i.e $D_{source}$ and $D_{target}$ each of which have examples as points in $\Re^{source}$ and $\Re^{target}$ respectively. The positive and negative classes of both the datasets can be assumed to be drawn from their corresponding probability distribution function, $P^+$ and $P^-$ respectively. From this point henceforth, we will be working with only the positive classes of both the datasets. Results for the negative class can be concluded in the same way.

## 2.3 ADVERSARIAL SETUP

Consider a deterministic function $G : \mathbb{R}^{source} \to \mathbb{R}^{target}$ parameterised by $\theta \in \Theta$, implemented by the means of a feed forward neural network. It takes a vector from the source domain and outputs a vector in the target domain. This function can be assumed to produce samples from a generating distribution, $P_g$. The goal is to minimize the KL divergence of these distributions and ideally making it zero, at which point, $P_{target} = P_g$.

This optimization can be achieved in an adversarial setting. Consider another function, $C : \mathbb{R}^{target} \to [0, 1]$ parameterised by $\pi \in \Pi$ also implemented by means of a fully connected network, albeit having a different architecture. This function outputs the probability that the input is from the original target data. The critic, $C$ tries to distinguish samples drawn from $P_g$ and $P_{source}$ whereas the generator, $G$ attempts to modify $P_g$ so as to fool the critic. The critic and the generator are trained alternatively using gradient descent.

This sets up the following minimax objective:

$$min_\pi max_\theta [E_{r\tilde{P}_{target}}[log(D_\pi(r))] - E_{z\ P_{source}}[log(D_\pi(G_\theta(z)))]]$$

## 3 EXPERIMENTS

We have tested our theoretical framework on two different datasets, viz. Street View House Numbers 1-7 and IMDB Sentiment Classification dataset.

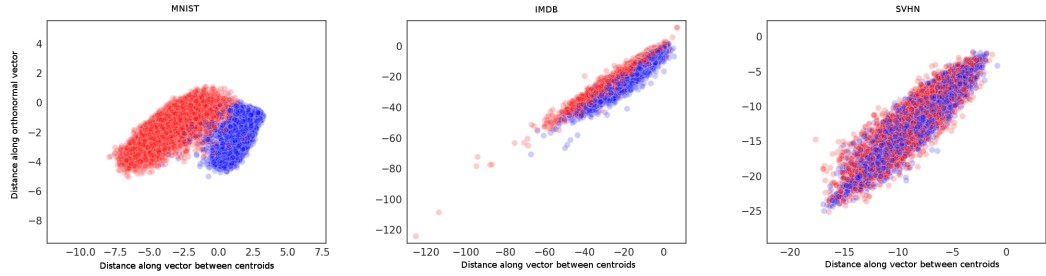

Figure 1: Plots of MNIST 1-7, IMDB and SVHN 1-7 datasets. The clusters are well-defined in the case of MNIST 1-7 but are not in the other two. SVHN 1-7 has the least amount of structure in the data.

Both critic and generator have been implemented by a 3 layer feed forward neural network.

In Fig 2 we see the plot of the datasets which are obtained after the adversarial transformation. There is an apparent class separability inherited from MNIST dataset which wasn't present in the source domain.

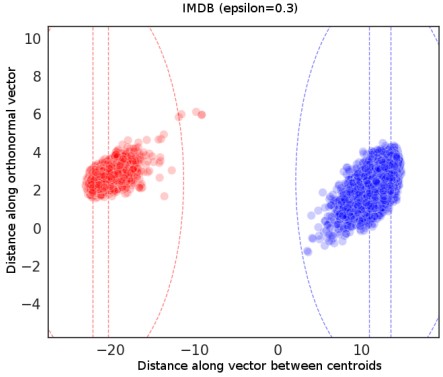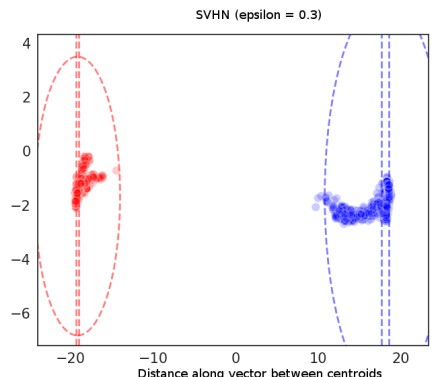

Figure 2: Plots of IMDB and SVHN dataset after the transformation. The intersection of the circle and slab denote the feasible region for that particular class as proposed.

Table 1: Upper bound on Test loss

| MNIST | IMDB | SVHN | IMDB Transformed | SVHN Transformed |
|---|---|---|---|---|
| 0.0455 | 1.0800 | 1.589 | $1.45 \times 10^{-6}$ | $2.08 \times 10^{-6}$ |

As the clusters obtained after transformation are at a significant distance from each other, the model is extremely resilient to attacks with the upper bound on test loss approaching 0. See Table 1. The upper bound is computed as proposed in (Steinhardt et al., 2017)

## 4 DISCUSSION

In this paper we provide a framework for improving the separability of the classes by adapting the representation of a dataset into another dataset which has an established separability. We have obtained high resilience to data attacks with upper bounds tending to zero.

## REFERENCES

Jacob Steinhardt, Pang Wei Koh, and Percy Liang. Certified defenses for data poisoning attacks. *CoRR*, abs/1706.03691, 2017. URL http://arxiv.org/abs/1706.03691.

