# OpenReview forum: "Domain Adversarial Representation Learning for Data Independent Defenses against Poisoning Attacks"
_ICLR.cc/2018/Workshop — Reject_

### Official Review · AnonReviewer3 · 2018-03-09
**Smart idea, but not developed enough**

**Rating:** 4
**Confidence:** 4

**Review:**

The paper build on recent work of Steinhardt et al. (2007) that proposed a method to make a learner more robust to "data poisoning attacks" by removing outliers. This method works under the assumption that classes are well-separated into clusters. The empirical study of Steinhardt et al. confirms this by showing great robustness on MNIST-17 dataset (where classes are well-separated) and poor robustness on IMDB dataset (where classes are not well separated).

In the current workshop papers, the authors start from the above empirical observation and undertake to transform the representation of IMDB samples to be alike the MNIST-17 ones. This is done by learning a mapping with the help of domain adversarial learning; they enforce the IMDB samples to be indistinguishable from the MNIST ones. Doing this, the experiments show this new representation makes the IMDB dataset more robust to attack under the Steinhardt et al. framework. I think this is interesting as MNIST and IMDB are in appearance unrelated tasks. (A similar result is obtained by transforming the SVHN dataset into the MNIST dataset, which are more related tasks).

The work focuses on a few experimental tasks. It shows that the idea is promising, but does not provide evidence that the method will work on other problems. Thus, the authors point out a nice observation, but the overall contribution is rather small.

Moreover, the paper lacks rigor: Few is said on how the method is trained, the objective function on page 2 contains undefined symbols (tilde{P}_{target}, D_\pi), and there is no reference to other works explaining the domain adversarial strategy.

---

### Official Review · AnonReviewer1 · 2018-03-10
**Overall, I could not get the motivation of the study.**

**Rating:** 3
**Confidence:** 3

**Review:**

The last sentence in Motivation is terminated in the middle.

What does R^{source}, R^{target} mean? Does the symbol "source" denote the number of the space dimension?

The construction of G in section 2.3 seems to follow the GAN. References to the original papers should be added. Also, no explanation on discriminator D is given. Is C is used as D in the formulation? In the formulation, pi is not defined.

Overall, I could not get the motivation of the study. There is no explanation how domain adaptation is used for what purposes. The description is so incomplete to understand the idea.

---

### Official Review · AnonReviewer2 · 2018-03-10
**Interesting but missed many details**

**Rating:** 5
**Confidence:** 4

**Review:**

[Overview]

This paper proposed a domain adaptation method based on adversarial learning, to alleviate the risk of poisoning attacks. Briefly, the authors proposed a learning method to transform the source domain to a target domain which has a more cleaner boundary among different classes. The learning method borrowed the idea from adversarial domain adaptation. It is shown that with the adversarial domain adaptation, the upper bound on test loss decrease significantly on IMDB and SVHN with MNIST as the target domain.

[Comments]

I think the only difference between the proposed objective function and the one of original GAN is that the input z to G_{\theta} is from a specific source domain, instead of a random noise, e.g., gaussian.

The paper missed many details. For example, what kind of representation was used to represent the source and target domain data. The architecture for generator and critic are also not explained. Also, the authors missed a number of references which should be relevant to this work.  In the experiment, the authors did not explain how to compute the upper bound on test loss.

I am also curious that what if the source domain data are replaced with two gaussian components, and how the upper bound on test loss change with the change of the multi-gaussian distributions.

I think this paper is not ready to be accepted but it would be a very interesting paper if the authors spend more effort to make the paper more reading-proof, explain the details in the experiments, and perform more ablated study in the experiments.

---

### Decision · Program_Chairs · 2018-03-20
**ICLR 2018 Workshop Acceptance Decision**

**Decision:**

Reject

**Comment:**

Based on the reviews, this paper has not been accepted for presentation at the ICLR workshop. However, the conversation and updates can continue to appear here on OpenReview.